# The Reactive Sites of Methane Activation: A Comparison of IrC_3_^+^ with PtC_3_^+^

**DOI:** 10.3390/molecules26196028

**Published:** 2021-10-04

**Authors:** Zizhuang Liu, Hechen Wu, Wei Li, Xiaonan Wu

**Affiliations:** 1Department of Chemistry, Fudan University, Shanghai 200433, China; 18210220047@fudan.edu.cn (Z.L.); hcwu19@fudan.edu.cn (H.W.); 2School of Mathematics and Physics, North China Electric Power University, Beinong Road 2, Huilongguan, Beijing 102206, China; weil@ncepu.edu.cn

**Keywords:** mthane activation, mass spectrometry, quantum chemical calculation, ractive site

## Abstract

The activation reactions of methane mediated by metal carbide ions *M*C_3_^+^ (*M* = Ir and Pt) were comparatively studied at room temperature using the techniques of mass spectrometry in conjunction with theoretical calculations. *M*C_3_^+^ (*M* = Ir and Pt) ions reacted with CH_4_ at room temperature forming *M*C_2_H_2_^+^/C_2_H_2_ and *M*C_4_H_2_^+^/H_2_ as the major products for both systems. Besides that, PtC_3_^+^ could abstract a hydrogen atom from CH_4_ to generate PtC_3_H^+^/CH_3_, while IrC_3_^+^ could not. Quantum chemical calculations showed that the *M*C_3_^+^ (*M* = Ir and Pt) ions have a linear M-C-C-C structure. The first C–H activation took place on the Ir atom for IrC_3_^+^. The terminal carbon atom was the reactive site for the first C–H bond activation of PtC_3_^+^, which was beneficial to generate PtC_3_H^+^/CH_3_. The orbitals of the different metal influence the selection of the reactive sites for methane activation, which results in the different reaction channels. This study investigates the molecular-level mechanisms of the reactive sites of methane activation.

## 1. Introduction

Methane has attracted attention as the main component of natural gas and the conversion of methane into value-added chemicals is very important [1,2]. However, methane is extremely stable, with high C–H bond strengths (439 kJ/mol), negligible electron affinity and low polarizability [3]. At present, most of the catalytic conversion of methane needs to be carried out under high-temperature or high-pressure conditions [4]. Metal carbides are a kind of molecules with the potential to activate methane and have been studied by several groups [5]. Research on the mechanisms of the activation of methane by metal carbides is of great value and it may be helpful to find new catalysts of metal carbides [6,7]. At the same time, it is very difficult to study the activation mechanisms of methane. In recent years, people have found that the study of the gas-phase reaction of methane is an important means to study related reaction mechanisms [8,9].

Past studies have shown that gas-phase clusters are an ideal model for studying the reaction mechanisms in condensed-phase systems. The study of gas-phase clusters can reveal the specific reaction mechanisms, including the active sites, and provide references for condensed-phase catalytical processes [10]. At present, researchers are concerned about the reactions of methane with metal ions such as Os^+^ [11], Pt^+^ [12], Ta^+^ [13] and Rh(0) [14] and metal oxides such as MgO^+^ [15], PbO^+^ [16], V_4_O_10_^+^ [17] and Re_2_O_7_^+^ [18]. In addition to metal ions and their oxides, studies have shown that metal carbides can also activate methane, such as FeC_6_^−^ [19], Mo_2_C_2_^−^ [20], Ta_2_C_4_^−^ [21], FeC_3_^−^ [6], AuC^+^ [22], FeC_4_^+^ [23] and *M*C^+^ [7]. These studies have explored some possible mechanisms for the activation of methane by metal carbide clusters. For example, the study of AuC^+^ reveals a special hydride-transfer mechanism (HT) [22], while the study of FeC_3_^-^ shows that methane and atomic clusters generate CC-coupling reaction products at high temperatures and explained the possible mechanism of non-oxidized methane aromatization at the molecular level [6]. The study of FeC_4_^+^ shows that the cluster can activate the C–H bonds of methane via the hydrogen-atom transfer (HAT) mechanism at ambient temperature; the study used the frontier orbital theory to explain the root cause of the HAT reaction [23]. Although there have been some studies on the mechanisms of metal carbides to activate methane in the past, there is still no clear investigation on the reactive sites of methane activation. Therefore, further research is needed to study the reactive sites for the activation of methane. Here, we reported the reactions of *M*C_3_^+^ (*M* = Pt and Ir) with methane.

## 2. Experimental and Computational Methods

The experiments were performed using an ion trap mass spectrometer equipped with a laser vaporization–supersonic expansion ion source that was reported previously [24,25]. The *M*C_3_^+^ (*M* = Pt and Ir) ions were generated by pulsed laser ablation of a rotating and translating metal/carbon (metal:carbon = 1:4) target. The nascent ablated plasma was entrained by a helium carrier gas with a backing pressure of about 0.5 MPa. The ions were mass-selected by a quadrupole and then were sent into a linear ion trap, where the ions were accumulated and cooled by helium gas. The *M*C_3_^+^ ions reacted with CH_4_, CD_4_ and ^13^CH_4_, introduced by a pulsed valve. After a 10 ms reaction, the trapped ions were ejected for mass detection.

Theoretical calculations were performed using the Gaussian 09 package [26]. All of the calculations were performed using the BMK functional with the def2-TZVP basis sets [27,28]. Vibrational frequency calculations were employed to identify the nature of reaction intermediates, transition states (TSs) and products. The Molclus program [29] was used to search for the possible stable structures of the *M*C_3_^+^, *M*C_2_H_2_^+^, *M*C_4_H_2_^+^ and *M*C_3_H^+^ (*M* = Pt and Ir). The low-lying stable isomers were then re-optimized at the BMK/def2TZVP level to confirm the relative energy sequence. Transition-state optimizations were performed with the synchronous transit-guided quasi-Newton (STQN) method and were verified through intrinsic reaction coordinate (IRC) calculations [30,31]. Vibrational frequency calculations were performed to identify the nature of reaction intermediates, transition states (TSs) and products.

## 3. Results and Discussion

The mass spectra for the reactions of mass-selected *M*C_3_^+^ (*M* = Pt and Ir) ions with He (a1 and a2), CH_4_ (b1 and b2), CD_4_ (c1 and c2) and ^13^CH_4_ (d1 and d2) in the ion trap at room temperature are shown in Figure 1. No product ion was observed in the mass spectra when using pure He as reactant gas, while two main peaks, at *m*/*z* = 219 and 243 for the Ir-system, and *m*/*z* = 222 and 246 for the Pt-system for the reactions with CH_4_, which can be attributed to the product ions with chemical formulas *M*C_2_H_2_^+^ and *M*C_4_H_2_^+^ (*M* = ^196^Pt and ^193^Ir), were observed to be the major reaction products. A weak mass peak at *m*/*z* = 233 assigned to the ion with chemical formula PtC_3_H^+^ for PtC_3_^+^/CH_4_ was also observed, while no IrC_3_H^+^ was found. The mass spectra suggest that two main reaction channels for both systems and one hydrogen-atom abstraction channel for PtC_3_^+^ were observed. The first channel is the formation of the *M*C_2_H_2_^+^ cation with the release of a neutral C_2_H_2_ (reaction 1). The second channel is the generation of the *M*C_4_H_2_^+^ ion with concomitant elimination of a dihydrogen molecule (reaction 2). The third channel is the generation of the PtC_3_H^+^ ion with the release of CH_3_ as shown in reaction 3.
*M*C_3_^+^ + CH_4_ → *M*C_2_H_2_^+^ + C_2_H_2_(1)
*M*C_3_^+^ + CH_4_ → *M*C_4_H_2_^+^ + H_2_(2)
PtC_3_^+^ + CH_4_ → PtC_3_H^+^ + CH_3_(3)

Impurity PtC_3_•H_2_O^+^ and IrCO^+^ ions were formed due to the small amount of contaminant in the chamber of the instrument. Isotopic-labeling experiments conducted using the CD_4_ sample showed that there were peaks of IrC_2_D_2_^+^, IrC_4_D_2_^+^, PtC_2_D_2_^+^, PtC_4_D_2_^+^ and PtC_3_D^+^, which demonstrates that the H atoms in the products were all from methane. Both the PtC_2_H_2_^+^ and Pt^13^CCH_2_^+^ product ions were observed to have approximately the same intensity, when using ^13^CH_4_, indicating that one or both carbon atoms of the eliminated C_2_H_2_ neutral molecule came from the PtC_3_^+^ ion. The intensity of the peak of Ir^13^CCH_2_^+^ is similar with that of IrC_2_H_2_^+^, which eliminates the background peak. The peaks of Ir^13^CC_3_H_2_^+^ and Pt^13^CC_3_H_2_^+^ were observed, which confirms the reaction channels.

In order to gain insight into the reaction mechanisms, the various possible structures of the products *M*C_3_^+^ (*M* = Pt and Ir) were obtained by calculations at the BMK/def2-TZVP level and are shown in Appendix A. The most stable structures of *M*C_3_^+^ (*M* = Pt and Ir) have a linear structure *M*-C-C-C, with the ground state ^1^Σ^+^ (IrC_3_^+^) and ^2^Σ^−^ (PtC_3_^+^). For the products (Appendix A), the most stable structures of *M*C_2_H_2_^+^ and *M*C_4_H_2_^+^ (*M* = Pt and Ir) are metal cations with C_2_H_2_ and linear H-C-C-C-C-H, respectively. For *M*C_3_H^+^ (*M* = Pt and Ir), the most stable structures of *M*C_3_H^+^ (*M* = Pt and Ir) have a linear structure *M*-C-C-C-H, given in Appendix A.

In order to gain insight into the reaction mechanism, the potential energy profiles (PESs) were calculated. All of the three Reactions (1)–(3) leading to the stable structures of the products were exothermic. The pathways for the IrC_3_^+^ + CH_4_ reaction leading to the IrC_2_H_2_^+^/C_2_H_2_ and IrC_4_H_2_^+^/H_2_ products are shown in Figure 2 and details are given in Appendix A. An encounter complex (*^3^**I1*), that is −1.51 eV lower in energy than the ground state reactants, is formed initially. The Ir atom serves as the active site of the reaction and intermediate *^1^**I2* is formed via the first C–H bond activation. Then, the CH_3_ moiety is transferred to C_3_ to form a C–C bond; meanwhile, one H atom is transferred to the Ir atom (*^1^I2* → *^1^**TS2* → *^1^**I3*). Subsequently, two hydrogen atoms transfer from the metal Ir atom to the C atom to form two new C–H bonds (*^1^I3* → *^1^**TS3* → *^1^**I4* → *^1^**TS4* → *^1^**I5*). The H atom of the CH_2_ moiety is activated and transferred to the Ir metal (*^1^I5* → *^1^**TS5* → *^1^**I6)*; then, the H atom migrates from the Ir metal to the C atom which is not coordinated to other H atoms (*^1^I6* → *^1^**TS6* → *^1^**I7)*. After the CC bond is cleft, the intermediate *^2^**I8* is formed, which involves two equivalent C_2_H_2_ moieties—either one can be liberated to form the final product ***P1*** (^3^IrC_2_H_2_^+^/C_2_H_2_). For the pathway for the generation of IrC_4_H_2_^+^/H_2_, there is the rearrangement steps from *^1^I3* to form intermediate *^1^I10* (*^1^I3* → *^1^**TS8* → *^1^**I9* → *^1^**TS9* → *^1^**I10*). The final product ^1^IrC_4_H_2_^+^ is generated with the liberation of H_2_. Another possible pathway is given in Appendix A.

The pathways for the PtC_3_^+^ + CH_4_ reaction are shown in Figure 3. Details and other possible pathways are given in Appendix A. The reaction pathway starts from the encounter complex *^2^I1*, followed by the formation of a stable intermediate *^2^I2* via the approaching of CH_4_ to the terminal carbon of PtC_3_^+^ (*^2^I1* → *^2^TS1* → *^2^I2*), where the first C–H activation takes place. The CH_3_ moiety of *^2^I2* is transferred to form *^2^I3*, where CH_3_ is loosely coordinated to PtC_3_H^+^. The final product *P1* (−0.46 eV) is generated with the liberation of CH_3_. The energy of *P1* is higher than the transition states from *^2^I2* to *P2/P3*, which can explain the weak peak of PtCH_3_^+^, compared with PtC_4_H_2_^+^ and PtC_2_H_2_^+^. From *^2^I2*, the H atom from the CH_3_ moiety is activated and transferred to the metal center to form intermediate *^2^I4* (^2^*I2*→ *^2^TS3* → *^2^I4)*. The H atom subsequently migrates from the Pt metal to the terminal C atom to form intermediate ^2^I5 (*^2^I4* → *^2^TS4* → *^2^I5)*. After rearrangement (*^2^I5* → *^2^TS5* → *^2^I6)*, the H atom of the CH_2_ moiety is activated and transferred to the Pt metal again (*^2^I6* → *^2^TS6* → *^2^I7)*. Then, the H atom migrates from the Pt metal to C atom which is not coordinated to other H atoms (*^2^I7* → *^2^TS7* → *^2^I8)*. After the C and Pt atoms are bonded, the stable intermediate *^2^I9* with a five-membered ring is generated. After the CC bond is cleft, the intermediate *^2^I10* is formed, which involves two equivalent C_2_H_2_ moieties—either one can be liberated to form the final product *P2* (^2^PtC_2_H_2_^+^/C_2_H_2_). For another pathway to generate PtC_4_H_2_^+^/H_2_, after the rearrangement from *^2^I5* to form *^2^I11*, the H atom of the CH_2_ moiety is activated and transferred to the Pt metal again (*^2^I11* → *^2^TS11* → *^2^I12)*. The subsequent reaction pathway is the H–H bond formation; then, the final product PtC_4_H_2_^+^ is generated with the liberation of H_2_.

For the reactions of the IrC_3_^+^ and PtC_3_^+^ with CH_4_, we could find the hydrogen abstraction reaction products PtC_3_H^+^ and no IrC_3_H^+^ was observed. Based on our calculations, the reactive sites play a key role. There are two possible sites, the metal and terminal carbon atom (the other two carbon atoms are fully bonded). For IrC_3_^+^/CH_4_, the barrier to the first C–H activation on the terminal C atom (+0.04 eV; see Appendix A) is higher in energy than *^1^TS1* (−0.68 eV). Based on the lack of IrC_3_H^+^ observed, we confirm that the reactive site of the first C–H activation is the Ir atom. For PtC_3_^+^/CH_4_, the energy required for the first C–H bond activation on Pt and C atoms is +0.01 (*^2^TSX3*) and +0.07 eV (*^2^TS1*), given in Figure 3 and Appendix A, respectively. Though the Pt atom, as a reactive site, is favorable, the transition state of H atom transfer as the next step (+0.19 eV, *^2^TSX4;*
Appendix A) is higher in energy than *^2^TS1*. Based on the observation of PtC_3_H^+^ in the experiment, we confirm that the terminal carbon atom can be a reactive site for PtC_3_^+^/CH_4_.

As we all know, metals have a stronger adsorption capacity for methane than non-metals. The different reactive sites can be explained by the analysis of the orbital. In the PtC_3_^+^ species, platinum uses two of the six valence orbitals (s and d) to form a σ- and π-bond with the adjacent carbon atom. This leaves seven electrons occupying the four remaining non-bonding orbitals on platinum, such that there are no empty orbitals on the metal. Therefore, it is hard for the C−H bond to donate electron density to the metal and the energy of the transition state *^2^TSX3* is a little higher than the *^2^TS1* in Figure 3. For the IrC_3_^+^ species, Ir uses two of the six valence orbitals (s and d) to form a bond with the adjacent carbon atom, which leaves six electrons occupying the four remaining non-bonding orbitals on the Ir atom, such that there are enough empty orbitals on the metal, which is beneficial for the C–H bond activation of CH_4_. This can explain why the barrier to the first C–H activation on the terminal C atom (+0.04 eV) is higher than that of *^1^TS1* (−0.68 eV). Our work demonstrates that the orbitals of the metal influence the selection of the reactive sites for the activation of methane. The different reactive sites result in different reaction channels.

In conclusion, the activation of methane mediated by *M*C_3_^+^ (*M* = Ir and Pt) were comparatively studied at room temperature by gas-phase experiments with theoretical calculations. Mass spectrometric studies on the reactions of the *M*C_3_^+^ (*M* = Ir and Pt) ions with CH_4_ show that two main reaction channels were observed. The first channel is the formation of the *M*C_2_H_2_^+^ cation with the release of neutral C_2_H_2_. The second channel is the generation of the *M*C_4_H_2_^+^ ion with concomitant elimination of a dihydrogen molecule. Besides that, PtC_3_H^+^ could be found in the experiments, while IrC_3_H^+^ could not. Quantum chemical calculations suggest that the *M*C_3_^+^ (*M* = Ir and Pt) ions have a linear *M*-C-C-C^+^ structure. The Ir atom is the reactive site for the reaction of IrC_3_^+^/CH_4_. The generation of the PtC_3_H^+^ and PESs can confirm that the terminal carbon atom is the reactive site for PtC_3_^+^/CH_4_. The orbitals of the metal influence the selection of the reactive sites for methane activation, which results in the different reaction channels. Our work is helpful for understanding reactive sites of methane activation.

## Figures and Tables

**Figure 1 molecules-26-06028-f001:**
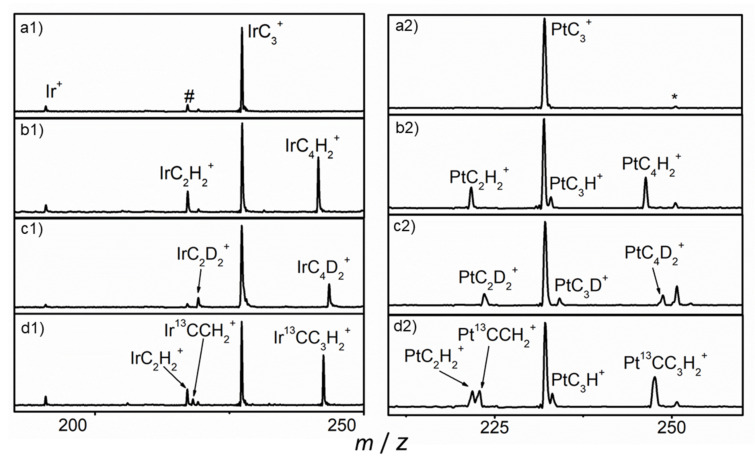
Mass spectra from the reactions of *M*C_3_^+^ (*M* = Pt and Ir) with He (**a1** and **a2**), CH_4_ (**b1** and **b2**), CD_4_ (**c1** and **c2**) and ^13^CH_4_ (**d1** and **d2**). # and * denote IrCO^+^ and PtC_3_H_2_O^+^.

**Figure 2 molecules-26-06028-f002:**
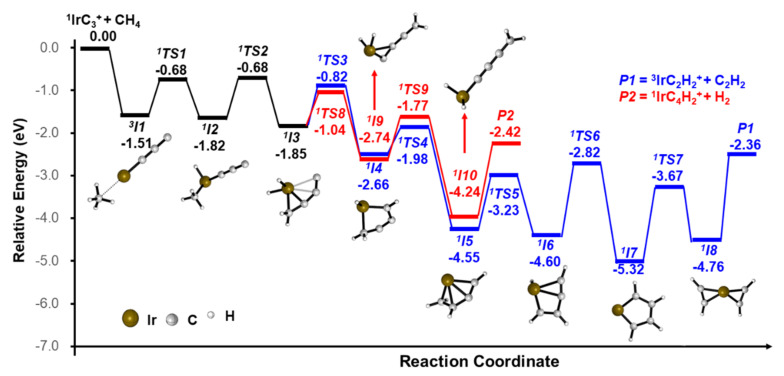
Reaction pathways for generating IrC_2_H_2_^+^/C_2_H_2_ (blue trace) and IrC_4_H_2_^+^/H_2_ (red trace) from the reaction of IrC_3_^+^ and CH_4_, calculated at the BMK level. The zero-point vibrational energy-corrected energies (in eV) relative to the entrance channel are given.

**Figure 3 molecules-26-06028-f003:**
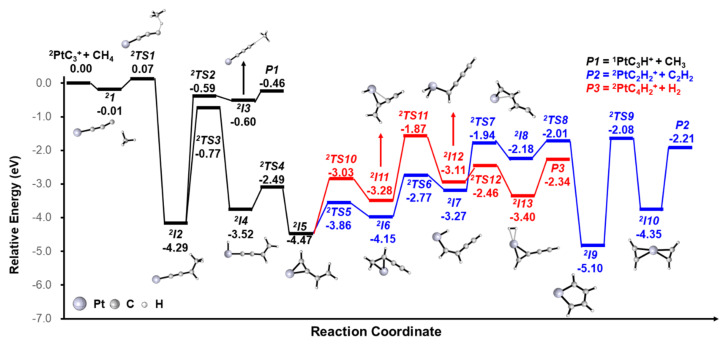
Reaction pathways for generating PtC_2_H_2_^+^/C_2_H_2_ (blue trace), PtC_4_H_2_^+^/H_2_ (red trace) and PtC_3_H^+^/CH_3_ from the reaction of PtC_3_^+^ and CH_4_, calculated at the BMK level. The zero-point vibrational energy-corrected energies (in eV) relative to the entrance channel are given.

## Data Availability

Not available.

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
