# Peer review of "The Reactive Sites of Methane Activation: A Comparison of IrC3+ with PtC3+"

_molecules, 2021, doi:10.3390/molecules26196028_

Round 1
Reviewer 1 Report
In reviewed work entitled “The Reactive Sites of Methane Activation: A Comparison of 2 IrC3+ with PtC3+” written by Zizhuang Liu, Hechen Wu, Wei Li and Xiaonan Wu the activation reactions of methane mediated by metal carbide ions (MC3+; M= Ir, Pt) have been studied. The mass spectrometry along with theoretical calculations have been engaged to examine this issue and determine the reactive sites in methane activation process. As an outcome the Authors exhibited that two main routes of reaction are present in investigated systems. First involving the MC2H2+ cation formation with release of neutral ethyne. The second generates MC4H2+ ion with accompanying elimination of dihydrogen bond. In my opinion this work represents concise and coherent vision of resolving research problem and therefore deserves to be publish in Molecules magazine. However, before publication I have some minor questions which need to be addressed by the Authors:
- Why the BMK functional was chosen? Were they any preliminary calculations or literature justification to select this particular functional?
- I wonder whether using Gibbs free energies instead of total electronic energies would change the results significantly. Did the Authors consider using this parameter?
- What method exactly was used to obtain transition states geometry? Namely, which keyword was used in input route: qst, qst2 or qst3?
Author Response
In reviewed work entitled “The Reactive Sites of Methane Activation: A Comparison of IrC3+ with PtC3+” written by Zizhuang Liu, Hechen Wu, Wei Li and Xiaonan Wu the activation reactions of methane mediated by metal carbide ions (MC3+; M= Ir, Pt) have been studied. The mass spectrometry along with theoretical calculations have been engaged to examine this issue and determine the reactive sites in methane activation process. As an outcome the Authors exhibited that two main routes of reaction are present in investigated systems. First involving the MC2H2+ cation formation with release of neutral ethyne. The second generates MC4H2+ ion with accompanying elimination of dihydrogen bond. In my opinion this work represents concise and coherent vision of resolving research problem and therefore deserves to be publish in Molecules magazine. However, before publication I have some minor questions which need to be addressed by the Authors:
1.Why the BMK functional was chosen? Were they any preliminary calculations or literature justification to select this particular functional?
Response: Thank you for the question. We have tested lots of functions in this paper, including the pure functionals BP86, M06-L, M11-L, B97D3, and TPSS; hybrid functionals with a varying amount of Hartree–Fock (HF) exchange, TPSSh (0.1), PBE1PBE (0.25), BMK (0.42), BH&HLYP (0.50), M06-2X (0.54), wB97 (1.0), and M11 (0.428/1.0); HF and MP2 methods. We found that only using BMK and few functions can get the stable intermediate PtC3H…CH3 (2I3) in Figure 3, which can explain the experimental results. Other functionals can’t not get the intermediate 2I3 in Figure 3. So we choose BMK functionals. The reference Goel, S.; Masunov, A. E. Theory. Int. J. Quantum. Chem. 2011, 111, 4276-4287 also use this functional to calculate the metal carbides.
2.I wonder whether using Gibbs free energies instead of total electronic energies would change the results significantly. Did the Authors consider using this parameter?
Response: We use the sum of electronic and zero-point Energies (ΔH), which is widely used in the PESs calculations. We also get the Gibbs free energies, but we don’t use here.
3.What method exactly was used to obtain transition states geometry? Namely, which keyword was used in input route: qst, qst2 or qst3?
Response: Our method use “ts” as the keyword. In predicting the reaction pathways, the intrinsic reaction coordinate (IRC) calculations were performed to confirm the correctness of the transition states
Reviewer 2 Report
Title: The Reactive Sites of Methane Activation: A Comparison of IrC3+ with PtC3+
by Zizhuang Liu, Hechen Wu, Wei Li and Xiaonan Wu
It is a nice scientific paper which explorers the activation reactions of methane induced by metal carbide ions. Theoretical and experimental parts are well described and sound credible. I suggest to publish the article on Molecules after minor revision:
1) Page 1, Introduction: Please change "," with "." after FeC4+
2) Did the authors try to use different theoretical methods? Why they choose BMK functional? It should be better clarify in the text.
3) Reference 26 should be updated with GAUSSIAN09
4) Please change the capital letters in reference 31
Author Response
It is a nice scientific paper which explorers the activation reactions of methane induced by metal carbide ions. Theoretical and experimental parts are well described and sound credible. I suggest to publish the article on Molecules after minor revision:
- Page 1, Introduction: Please change "," with "." after FeC4+
Response: we have changed it.
- Did the authors try to use different theoretical methods? Why they choose BMK functional? It should be better clarify in the text.
Response: Thank you for the question. We have tested lots of functions in this paper, including the pure functionals BP86, M06-L, M11-L, B97D3, and TPSS; hybrid functionals with a varying amount of Hartree–Fock (HF) exchange, TPSSh (0.1), PBE1PBE (0.25), BMK (0.42), BH&HLYP (0.50), M06-2X (0.54), wB97 (1.0), and M11 (0.428/1.0); HF and MP2 methods. We found that only using BMK and few functions can get the stable intermediate PtC3H…CH3 (2I3) in Figure 3, which can explain the experimental results. Other functionals can’t not get the intermediate 2I3 in Figure 3. So we choose BMK functionals. The reference Goel, S.; Masunov, A. E. Theory. Int. J. Quantum. Chem. 2011, 111, 4276-4287 also use this functional to calculate the metal carbides.
- Reference 26 should be updated with GAUSSIAN09
Response: we have changed it. Thank you.
4) Please change the capital letters in reference 31
Response: we have changed it.